# Heliconical smectic phases formed by achiral molecules

Jordan P. Abberley[1], Ross Killah[1], Rebecca Walker[1], John M.D. Storey[1], Corrie T. Imrie[1], Mirosław Salamończyk [2,3], Chenhui Zhu[2], Ewa Gorecka[4] & Damian Pociecha[4]

Chiral symmetry breaking in soft matter is a hot topic of current research. Recently, such a phenomenon was found in a fluidic phase showing orientational order of molecules— the nematic phase; although built of achiral molecules, the phase can exhibit structural chirality—average molecular direction follows a short-pitch helix. Here, we report a series of achiral asymmetric dimers with an odd number of atoms in the spacer, which form twisted structures in nematic as well as in lamellar phases. The tight pitch heliconical nematic ($N_{TB}$) phase and heliconical tilted smectic C ($SmC_{TB}$) phase are formed. The formation of a variety of helical structures is accompanied by a gradual freezing of molecular rotation. In the lowest temperature smectic phase, HexI, the twist is expressed through the formation of hierarchical structure: nanoscale helices and mesoscopic helical filaments. The short-pitch helical structure in the smectic phases is confirmed by resonant X-ray measurements.

[1] Department of Chemistry, King's College, University of Aberdeen, Aberdeen AB24 3UE, UK. [2] Advanced Light Source, Lawrence Berkeley National Laboratory, 1 Cyclotron Road, Berkeley, CA 94720, USA. [3] Department of Physics and Liquid Crystal Institute, Kent State University, 1425 Lefton Esplanade, Kent, OH 44242, USA. [4] Faculty of Chemistry, University of Warsaw, Zwirki i Wigury 101, 02-089 Warsaw Poland. Correspondence and requests for materials should be addressed to D.P. (email: pociu@chem.uw.edu.pl)

The spontaneous formation of chiral structures from effectively achiral molecules is at the heart of intense worldwide research, as the breaking of mirror symmetry is a fundamental issue in chemistry, physics and biology and plays a central role in the origin of biological homochirality. Twisted structures made from chiral building blocks are relatively common in nature, the archetypal examples being DNA or proteins that can exist in helical forms not only in solids but also in the liquid state, whereas examples of achiral molecules that assemble into helical aggregates are much less common, and until recently such helical aggregates were observed only for 3D ordered crystals. In the area of liquid crystals (LC), interest in chiral phases consisting of achiral building blocks began with the report by Sekine et al.[1] describing the properties of so-called bent-core or banana LCs. Although it was to transpire that the new phase (later designated $B_4$) reported in this work and described as a low dimensional LC system, was in fact a 3D crystal. However, the intense research stimulated by the Tokyo group's finding ultimately resulted in the discovery of the twist-bend heliconical nematic ($N_{TB}$) phase[2, 3], some 35 years after its prediction by Meyer[4]. In the intervening period, Dozov independently predicted the existence of the helical nematic and smectic phases using symmetry arguments[5]. At the root of Dozov's prediction was the assumption that bent molecules have a natural tendency to pack into bent structures, however as uniform bend is not permitted in nature it must be accompanied by other local deformations, namely either splay or twist of the average molecular axis direction (director). The splay-bend nematic phase is achiral, by contrast, in the $N_{TB}$ phase the director forms a conical left- or right-handed helix. The $N_{TB}$ phase provided the first example of spontaneous chiral symmetry breaking in a fluid with no spatial ordering[2, 3, 6–8]. The helix in the $N_{TB}$ phase is extremely short, typically ~ 10 nm (3–4 molecular distances)[6, 7, 9, 10]. For the overwhelming majority of twist-bend nematogens, the $N_{TB}$ phase is preceded by a conventional nematic (N) phase with uniform director structure, for which the strongly bent molecules give rise to small values of the bend elastic constant[8]. In fact, there are just three examples of direct $N_{TB}$—isotropic phase transitions[11–13]. On cooling, the vast majority of $N_{TB}$ phases either crystallise or vitrify, and only rarely is a $N_{TB}$-smectic phase transition observed, specifically either a smectic A phase typical for rod-like molecules (see, for example[14, 15]), or a broken layer modulated $B_1$ type phase[16], typical for bent molecules have been observed. We have yet to establish and understand how the bent molecules will self-assemble into smectic phases if the bend elastic constant becomes anomalously low. Here we show that as in the nematic phase, such achiral bent molecules also spontaneously form short-pitch length helical structures in smectic phases.

The simplicity of the twist-bend nematic phase has significant application potential, and a conventional nematic phase having a small bend elastic constant may itself be utilised in new technological applications such as the electrically controlled selective reflection of light[17] and in an electrically tunable laser[18].

The molecular curvature, that is essential for the formation of the $N_{TB}$ phase, can be realised using odd-membered mesogenic dimers. Such dimers are built of two rigid mesogenic moieties linked by a flexible spacer, normally an alkyl chain[19]. The liquid crystalline behaviour of dimers is strongly dependent on the length and parity of the spacer and this is most commonly attributed to how the average shape of the molecule is controlled by conformations of the spacer. Thus, if we consider the spacer to exist in an all-trans conformation then for an even-membered spacer the long axes of the mesogenic groups are essentially parallel and the molecule is linear, whereas for an odd-membered spacer they are inclined at some angle with respect to each other giving a bent molecular shape.

Here, we report the characterisation of a homologous series of dimeric mesogenic molecules, which owing to the odd-membered linkage adopt bent geometry. For the short homologues the phase transition between the uniform and heliconical nematic phases was found, whereas for longer homologues, a complex sequence of smectic phases was observed in which the tilted phases show a helical precession of the molecules around the cone axis, similar to chiral smectic phases. The lowest temperature smectic phase was found to exhibit structural chirality at different length-scales, i.e., layer chirality, nanoscale helices and mesoscopic helical filaments.

## Results

**Phase behaviour.** The molecular structure of the 4-[{[4-({6-[4-(4-cyanophenyl)phenyl]hexyl}oxy)phenyl] methylidene}amino]phenyl-4-alkoxy-benzoates (CB6OIBeOn, in which $n$ represents the number of carbon atoms in the terminal alkyl chain) is shown in Fig. 1.

The hexyloxy spacer was chosen as it provides the molecular curvature required to observe the $N_{TB}$ phase, whereas increasing the terminal chain length should promote smectic phases. The studied dimers show a complex polymorphism of liquid crystalline phases, their phase sequences, transition temperatures and associated enthalpy changes are collected in Table 1, and the phase diagram based on calorimetric, X-ray diffraction (XRD) and optical studies is presented in Fig. 2a.

**X-ray studies.** Homologues with short terminal chains, $n = 1–6$, exhibit two nematic phases, N and $N_{TB}$, and a lamellar phase below the $N_{TB}$ phase. Both nematic phases give similar XRD patterns, evidencing only short-range positional ordering of the molecules (Supplementary Figure 1). For the N and $N_{TB}$ phases only very weak low-angle signals were seen in the X-ray patterns, corresponding to about the full and half molecular length, indicating a local short-range lamellar structure (cybotactic groups). For longer homologues, the $N_{TB}$ phase is extinguished, and instead, up to four lamellar phases were found below the conventional nematic phase. The layer spacing in the smectic phases corresponds approximately to the full molecular length. The lowest temperature lamellar phase is the same for both, short and

**Fig. 1** Molecular structure. General molecular structure of studied homologous series CB6OIBeOn

**Table 1 Phase sequence for studied compounds**

| n | m.p. | HexI | SmC$_{TB}$ | SmA$_b$ | SmA | N$_{TB}$ | N | Iso |
|---|------|------|-----------|---------|-----|----------|---|-----|
| 1 | 138.7 (48.3) | | | | | ● | 144.5 [a] ● | 283.5 (2.7) ● |
| 2 | 123.6 (52.2) | ● | 56.6 (4.29) | | | ● | 141.0 [a] ● | 279.4 (3.2) ● |
| 3 | 122.3 (36.8) | | | | | ● | 135.7 [a] ● | 267.6 (2.2) ● |
| 4 | 131.1 (47.3) | ● | 66.2 (5.1) | | | ● | 132.0 [a] ● | 260.0 (2.1) ● |
| 5 | 127.3 (39.9) | ● | 71.5 (5.1) | | | ● | 129.8 [a] ● | 248.8 (1.9) ● |
| 6 | 102.7 (40.4) | ● | 73.6 (4.5) | | | ● | 127.0 [a] ● | 243.6 (1.7) ● |
| 7 | 115.4 (54.4) | ● | 84.8 (4.6) ● | 104.2 [b] ● | 104.4 (0.07) [b] ● | 122.8 (0.04) | ● | 236.5 (1.8) ● |
| 8 | 112.2 (49.6) | ● | 88.8 (4.3) ● | 100.4 [b] ● | 102.6 (0.06) [b] ● | 143.9 (0.05) | ● | 231.9 (2.0) ● |
| 9 | 112.0 (50.4) | ● | 89.9 (4.6) | | 99.3 (0.07) ● | 158.9 (0.09) | ● | 224.0 (1.6) ● |
| 10 | 101.4 (56.9) | ● | 94.5 (5.6) | | 99.4 (0.02) ● | 172.3 (0.11) | ● | 222.3 (1.5) ● |

Phase transition temperatures (°C) and associated enthalpy changes (in parentheses, kJ mol$^{-1}$) obtained by DSC for the CB6OIBeOn series. General molecular structure of CB6OIBeOn is also given
[a] From microscopic observation
[b] SmA–SmA$_b$ and SmA$_b$–SmC$_{TB}$ transitions were not resolved in the thermograms, combined enthalpy changes are given

long, homologues, its X-ray pattern in the high-angle range shows a narrowed, split signal.

The correlation length of the in-plane positional order, calculated from the width of the high-angle diffraction signal, corresponds to several molecular distances, suggesting a hexatic-type smectic phase[20]. The characteristic XRD pattern obtained for an aligned sample, in which one of the high-angle signals is seen in an equatorial position with respect to the low-angle signals (Supplementary Figure 2), shows that the molecules are tilted toward the apex of the local in-plane hexagons, and hence, the phase is assigned as a HexI phase[21]. For all the other smectic phases (the higher temperature smectics), XRD patterns with a single, broad high-angle signal were detected, indicating liquid-like positional ordering of the molecules within the smectic layers (Supplementary Figure 2). The phase transitions between the smectic phases are clearly visible as changes in the layer thickness measured versus temperature (Fig. 2b). Two of the liquid-like smectic phases, appearing in the sequence below the nematic phase, differ only in their values of the layer thermal expansion coefficient; for example, −0.03 Å K$^{-1}$ and −0.07 Å K$^{-1}$ for CB6OIBeO8. At the phase transition to the lowest temperature liquid-like smectic phase, a continuous decrease of layer spacing is observed, suggesting a transition to a tilted phase (Fig. 2b and Supplementary Figure 3). Finally, a jump in the layer thickness is seen at the transition to the HexI phase. Summarising, the X-ray studies allowed for the identification of the lamellar phases as two orthogonal smectic A variants, a tilted smectic C and a hexatic smectic I phase.

**Optical studies.** In optical studies the N–N$_{TB}$ phase transition is observed as a change in the microscopic texture. On decreasing the temperature to a few degrees below the transition to the N$_{TB}$ phase, in cells with planar anchoring the uniform texture seen for the conventional nematic phase is replaced by stripes with periodicity equal to the cell thickness (Fig. 3a). Interestingly, for the materials studied here a similar texture change is also observed for longer homologues at the transition to the tilted smectic (SmC) phase (Fig. 3b). The striped texture is thought to indicate a low bend elastic constant in the N$_{TB}$ phase and the observation of a similar texture in the tilted smectic phase strongly suggests that its bend elastic constant is also very low[22]. The stripes, formed in the N$_{TB}$ or smectic C phase, persist also in the HexI phase (Fig. 3c) to the temperature at which the material crystallises. The optical textures were also studied in cells with homeotropic anchoring and several other observations were noted; the highest temperature smectic phase is optically uniaxial, consistent with a SmA phase assignment; for the lower temperature orthogonal phase a schlieren-like texture is observed, and in the smectic C phase the texture again becomes uniformly black when observed between crossed polarizers (Fig. 2c).

The schlieren texture of the orthogonal smectic phase suggests that the phase is optically biaxial (SmA$_b$ phase), in which molecular rotation around the long axis is to some degree frozen. The homeotropic optical texture of the tilted smectic C phase excludes its simple (uniplanar) synclinic or anticlinic lamellar structure, as both SmC$_s$ and SmC$_a$ phases are biaxial and form schlieren textures. The optical uniaxiality of the tilted phase suggests an averaging of molecular orientations owing to the formation of a short helix—we term the phase the twist-bend SmC$_{TB}$ (Fig. 2d). The assumption is supported by the temperature dependence of the average optical birefringence, for which a decrease in Δn is observed at the SmA–SmC$_{TB}$ phase transition (Fig. 4a). Also precise determination of the birefringence across the stripe area shows the decrease of birefringence in the SmC$_{TB}$ phase with respect to the value measured in the smectic A phase (Fig. 4c). The conical tilt angle, θ, calculated from the decrease of Δn in the SmC$_{TB}$ phase (for details see Methods section and[23]) for CB6OIBeO7 saturates at ~ 10 degree, and the same value of the tilt angle has been deduced from changes of the smectic layer spacing (Fig. 4b).

The X-ray studies point toward a weakly first order transition from the orthogonal smectic to tilted smectic phase for homologue n = 7 and a continuous transition for homologue n = 8, with a critical exponent close to 0.4 (Supplementary Figure 3). For the N$_{TB}$ phase of homologue CB6OIBeO6, the conical angle, θ, obtained from the decrease of birefringence measured below the N–N$_{TB}$ phase transition is ~ 20 degree.

**RSoXS studies.** The suggestion that the smectic C phase is heliconical is further supported by results obtained using soft X-ray resonant scattering (RSoXS) at the carbon absorption K-edge, a technique used previously to probe the helical structure of the N$_{TB}$ phase[10] and twisted filament structure of the B$_4$ phase[24]. On cooling, at the SmA–SmC$_{TB}$ phase transition a sharp, resonant Bragg peak develops, corresponding to the pitch length p ≈ 150 Å (Fig. 4d). The pitch slightly shortens with decreasing temperature, being incommensurate with layer structure, and corresponds to 3−4 smectic layer distances. The resonant signal has been detected also in the hexatic smectic phase, in this phase the peak position shows only weak temperature dependence (it changes in the range 100−102 Å and corresponds to ~ 2.2 smectic layer distances). The SmC$_{TB}$-HexI transition is first order with a clear discontinuity of the superlayer structure and a small temperature range in which both phases coexist (Supplementary Figure 4).

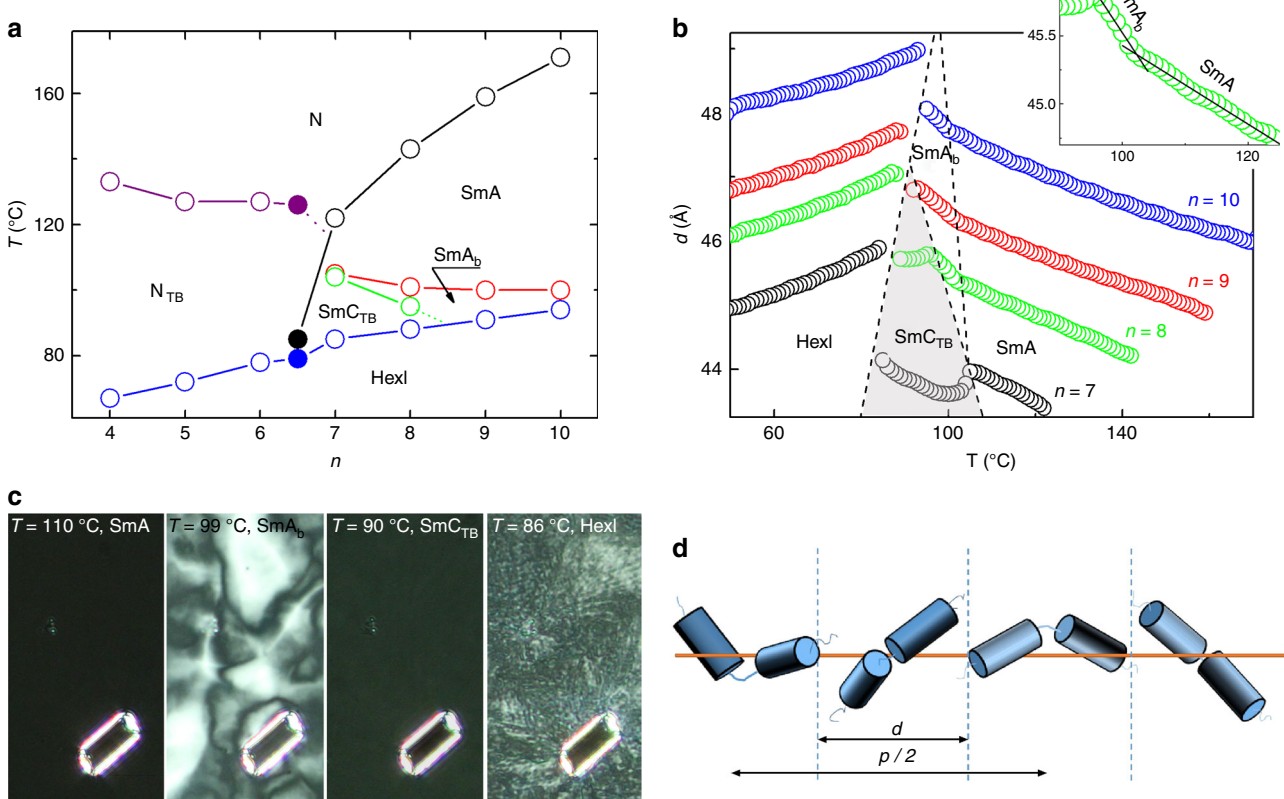

**Fig. 2** Phase behaviour. **a** Phase diagram for the achiral CB6OIBeOn series with $n = 4$–10, the filled points correspond to a 50–50 wt.% mixture of homologues $n = 6$ and 7. **b** Smectic layer spacing, $d$, vs. temperature for CB6OIBeOn compounds with $n = 7$–10. The dependence, $d(T)$, for $n = 7$ and 8 suggests a transition from an orthogonal to a tilted phase. The highlighted area is the heliconical smectic $C_{TB}$ phase. In the inset is the enlarged fragment of $d$ vs. $T$ for homologue $n = 8$, which clearly shows the SmA-SmA$_b$ phase transition. **c** Optical textures of the uniaxial SmA, biaxial SmA$_b$, helical SmC$_{TB}$ and HexI phases, observed in an homeotropic cell, between crossed polarizers, for homologue CB6OIBeO8. A glass bead is visible in the picture to mark the same area of the presented textures. Note that the SmC$_{TB}$ phase is uniaxial, which excludes a simple synclinic or anticlinic tilted structure. **d** Schematic drawing of the proposed model for SmC$_{TB}$ phase structure in which dimeric molecules form a short-pitch helix, $p$ stands for pitch length. The dotted vertical lines denote smectic layer interfaces, spaced by $d$

The texture of the HexI phase in cells with homeotropic anchoring is very weakly birefringent (Fig. 1c), and a slight de-crossing of the polarizers reveals the presence of optically active domains (Supplementary Figure 6). The HexI phase gives a clear circular dichroism signal around the absorption band of the material (~ 350 nm), consistent with the chiral nature of the phase (Supplementary Figure 7). The morphology of the sample in the HexI phase was studied by the AFM method. For the samples aligned between glass slides with planar anchoring, large areas with uniformly oriented entangled filaments, with an average diameter of ~ 50 nm, were found (Fig. 3d and Supplementary Figure 8). The filaments appear to have uniform twist sense over micron size areas. The morphology of the sample strongly resembles that observed for the B$_4$ phase, although it should be noted that filaments of B$_4$ phase have an internal crystalline structure[25, 26]. The origin of chiral optical effects in the HexI phase is not clear; for tilted smectics consisting of chiral molecules, it is known that optical activity becomes negligible if the pitch is much shorter than the wavelength of light. Therefore, optical activity and CD in the HexI phase should be attributed, as with the B$_4$ phase, to the internal layer structure (the 'layer chirality') induced by the strongly non-homogenous electron distribution across the layer and the tilted arrangement of the strongly biaxial molecules[25, 27]. Lack of CD and optical activity in the SmC$_{TB}$ phase indicates that essentially free molecular rotation

around the molecular long axis occurs, and therefore the biaxiality is too weak to cause detectable 'layer chirality'.

No clear optical flexoelectric switching, reported previously for other N$_{TB}$ materials, was found for the studied compounds. Instead, only a weak change in the birefringence and a bi-polar response under high electric fields were detected, except for the HexI phase, which was not sensitive to electric field.

## Discussion

Summarising, for the short homologues of the CB6OIBeOn series the phase transition between the uniform N and heliconical N$_{TB}$ phases was found, whereas for longer homologues, as the tendency for the alkyl chains and aromatic moieties to nanoscale separate increases, a complex sequence of smectic phases was observed in which the tilted phases show a helical precession of the molecules around the cone axis, similar to chiral smectic phases. The heliconical smectic phase is preceded by the non-tilted SmA and SmA$_b$ phases. Apparently, on lowering the temperature rotation around the long molecular axis is gradually frozen and as a result the transition from the SmA to SmA$_b$ phase is observed. The gradual freezing of rotation is accompanied by a more pronounced layer spacing increase, owing to lower inter-digitation of molecules between neighbouring layers and a stretching of the terminal chains. The lack of a polar response

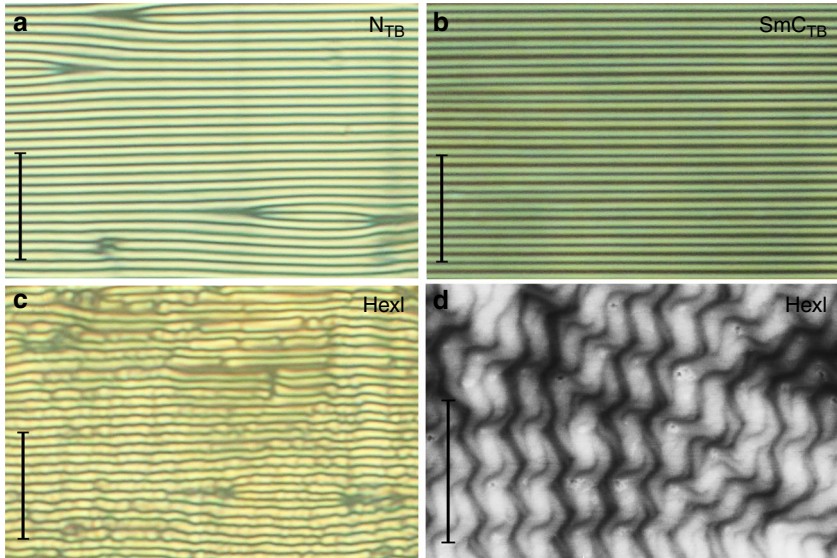

**Fig. 3** Optical textures. Stripe textures observed in the 1.9 micron cells with planar anchoring for **a** $N_{TB}$ phase of CB6OIBeO6, **b** $SmC_{TB}$ phase of CB6OIBeO7 and **c** HexI phase of CB6OIBeO6. **d** AFM image of twisted entangled filaments formed in the temperature range of the HexI phase for homologue CB6OIBeO7. Scale bars in **a**–**c** correspond to 20 μm, whereas in **d** to 500 nm

under an electric field shows that in the $SmA_b$ phase the dipole moments of the molecules are locally compensated. The X-ray studies revealed that upon lowering temperature, for intermediate length homologues the $SmA_b$ phase undergoes a transition (weakly first order or continuous) to a tilted phase, which surprisingly is optically uniaxial. This optical uniaxiality is inconsistent with a simple synclinic or anticlinic SmC phase structure, and indicates that averaging of molecular orientation must take place through the formation of a heliconical structure. The resonant X-ray scattering experiment indeed proves the existence of the superstructure, with a pitch corresponding to 3–4 smectic layers. This is the first report showing unambiguously that a short-pitch helical structure may be formed spontaneously in smectic phases consisting of achiral molecules. Such short helices were found previously for smectic phases comprising chiral rod-like molecules ($SmC_\alpha$ phase) as a result of competing interactions between nearest and next nearest layers—the frustration in systems, in which next nearest interactions favour antiparallel tilt, is relieved by the formation of a helix that can be as short as a few smectic layers[28, 29]. However the molecules studied here are achiral dimers and the helix formation is presumably driven by steric interactions arising from the bent molecular shape, in similar fashion to the $N_{TB}$ phase. The reported bent dimers have an extremely strong tendency to form twisted structures, this is evidenced not only by the heliconical arrangement of the molecules in the $N_{TB}$ and $SmC_{TB}$ phases, but also is expressed through formation of mesoscopic helical filaments in the lowest temperature smectic phase, the HexI phase.

## Methods

**Calorimetry**. Calorimetric studies were performed using differential scanning calorimeter (DSC) TA Q200, samples of mass 1–3 mg were sealed in aluminium pans and kept in nitrogen atmosphere during measurement, both heating and cooling scans were performed with rate 10 K min$^{-1}$.

**XRD**. Wide angle X-ray diffractograms were obtained with a Bruker D8 GADDS system (CuKα line, Goebel mirror, point beam collimator, Vantec2000 area detector). Samples were prepared as droplets on a heated surface. The temperature dependence of the layer thickness was determined from small-angle XRD experiments performed with a Bruker D8 Discover system (CuKα line, Goebel mirror, Anton Paar DCS350 heating stage, scintillation counter) working in reflection mode, homeotropically aligned samples were prepared as thin films on a silicon wafer. The tilt angle in the tilted smectic phase was calculated from the decrease of

the layer spacing, $d_{SmC} = d_{SmA} \cos(\theta)$, with respect to the layer thickness value extrapolated from the data recorded in the orthogonal SmA phase to the temperature range of the tilted phase.

The resonant X-ray experiment was performed on the soft X-ray scattering beamline (11.0.1.2) at the Advanced Light Source of Lawrence Berkeley National Laboratory. The X-ray beam was tuned to the K-edge of carbon strongest absorption peak, ~ 283.6 eV, however for the studied dimeric material a few more peaks with different energy were also detected. An energy scan was done at two temperatures: 392 and 360 K in the $SmC_{TB}$ and HexI phases, respectively. The X-ray beam with a cross-section of $300 \times 200$ μm was linearly polarised, with the polarisation direction that can be continuously changed from the horizontal to vertical. Samples with thickness lower than 1μm were placed between two 100-nm-thick $Si_3N_4$ slides. The scattering intensity was recorded using the Princeton PI-MTE CCD detector, cooled to −45 °C, having a pixel size of 27 μm, with an adjustable distance from the sample. The detector was translated off axis to enable a recording of the diffracted X-ray intensity. The adjustable position of the detector allowed us to cover a broad range of $q$ vectors, corresponding to periodicities from ~ 5.0–300 nm.

**Optical microscopy**. Optical studies were performed using the Zeiss Imager A2m polarising microscope equipped with Linkam heating stage. Samples were observed in glass cells with various thickness: 1.8–10 microns. The microscope setup was equipped with an Abrio system for precise birefringence measurements. The birefringence was calculated from optical retardation of green light ($\lambda = 546$ nm). The average retardation was measured in $10 \times 10$ micron region, but space modulation of retardation was also evaluated with submicron resolution. For birefringence measurements the three micron cells were used with planar alignment layer. As the system allows the measurement of retardation only up to 273 nm, the absolute value of retardation was determined therefore by comparing the results obtained for the samples having different thicknesses, 1.8 and 5 micron. The conical tilt angle in the $SmC_{TB}$ phase was deduced from the decrease of birefringence with respect to the values measured in the SmA phase, $\Delta n_{SmCTB} = \Delta n_{SmA}(3\cos^2(\theta)-1)/2$[23], the birefringence of the SmA phase was extrapolated to the lower temperature range, assuming a linear temperature dependence of $\Delta n$ in the SmA phase, the increase of the birefringence in the SmA phase on lowering temperature is due to the growing positional and/or conformational ordering of the molecules.

Electrooptic response of the phases was studied in glass cells (1.6–5.0 μm thick) covered with transparent ITO electrodes, and a surfactant layer ensuring planar sample alignment. Electric field of low frequency (<30 Hz) and magnitude up to 30 V μm$^{-1}$ was applied using. Light transmission was monitored with PIN-20 photodiode mounted in microscope.

**Atomic force microscopy**. AFM images have been recorded using a Bruker Dimension Icon microscope, working in tapping mode at the liquid crystalline-air surface. Cantilevers with a low spring constant, $k = 0.4$ Nm$^{-1}$ were used, the resonant frequency was in the range of 70–80 kHz, and the typical scan frequency was 1 Hz.

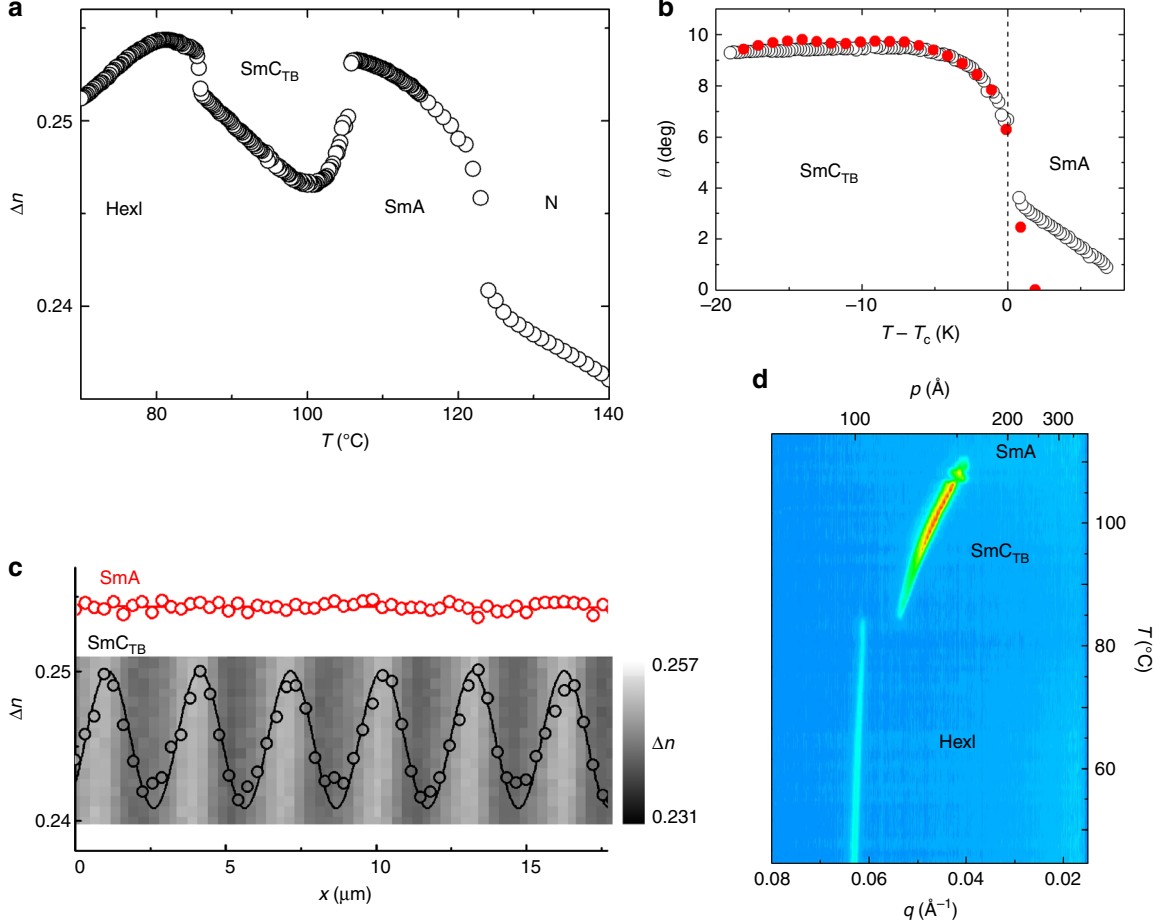

**Fig. 4** Birefringence and helical pitch. **a** Temperature dependence of the optical birefringence, $\Delta n$, for homologue CB6OIBeO7 measured in 3 μm-thick cell with planar anchoring. **b** Dependence of the conical angle $\theta$ deduced from both birefringence (open circles) and X-ray measurements (filled, red circles). **c** Changes of birefringence across the stripes shown in the background, revealing a decrease of $\Delta n$ in the $SmC_{TB}$ phase with respect to the orthogonal SmA phase owing to heliconical structure. **d** Temperature evolution of the resonant soft X-ray diffraction signal for the CB6OIBeO7 compound measured on cooling; whereas in the HexI phase the signal position is practically temperature independent (it corresponds to ~ 100 Å), in the $SmC_{TB}$ phase it changes strongly owing to the changes of the heliconical pitch length from 117 to ~ 150 Å close to the transition to the SmA phase. Note that the pitch dependence is monotonic, whereas the smectic layer spacing is a non-monotonic function of temperature

**Circular dichroism**. The CD spectra were collected with a J-815 spectropolarimeter from Jasco, Japan, using quartz plates as substrates for the samples.

**Data availability**. The data that support the findings of this study are available from the corresponding author upon reasonable request.

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

## Acknowledgements

M.S. acknowledges the support of the US National Science Foundation I2CAM International Materials Institute Award, Grant DMR-1411344 and NSF grant DMR-1307674. D.P., E.G. acknowledges the support of the National Science Centre (Poland) under the grant no. 2016/22/A/ST5/00319. R.W. gratefully acknowledges the Carnegie Trust for the Universities of Scotland for the award of a PhD studentship. The beamline 11.0.1.2 at the Advanced Light Source at the Lawrence Berkeley National Laboratory is supported by the Director of the Office of Science, Office of Basic Energy Sciences, of the U.S. Department of Energy under Contract No. DE-AC02- 05CH11231.

## Author contributions

J.P.A., R.W., J.M.D.S., C.T.I. were responsible for organic synthesis and calorimetric sudies, M.S., C.Z. were responsible for resonant X-ray studies, E.G., D.P. and R.K. performed optical, AFM and X-ray studies.

## Additional information

**Competing interests:** The authors declare no competing financial interests.

