## [Peer Review File · Nature Communications]

Reviewers' comments:

Reviewer #1 (Remarks to the Author):

The paper reports the synthesis and detailed characterization of a series of materials with dimeric molecules consisting of two rigid parts linked by a flexible spacer. Several liquid crystal phases are observed and characterized by microscopy, x-ray diffraction, circular dichroism and AFM. There are several interesting aspects of the phases formed. First there are examples of the recently established twist-bend nematic phase. Second there are several smectic phases, including a biaxial smectic A and a smectic C phase as well as a hexatic smectic I. The smectic C phase is the jewel in the crown. Here, it is formed from achiral molecules but careful optical and X-ray measurements have established that it is helicoidal. Thus it is a rare example of a chiral structure being formed from achiral constituent molecules. This is the main claim of the paper and is a significant observation that will be of widespread interest.

The helicoidal structure is strongly suggested by a combination of observations. That is the homeotropic optical texture for a smectic phase that has been shown to be tilted (via normal X-ray diffraction study of the phase diagram). It is then proved beyond reasonable doubt by resonance X-ray diffraction to observe the helix peaks. Very elegant.

The case is further strengthened by circular dichroism studies. The chiral domains turn out to be too small in the SmC but it is clearly observed in the hex-I phase.

Overall, the paper is very nicely presented with strong evidence to back up the conclusions. The level of detail is sufficient to allow other researchers in the field to reproduce the work. There are several interesting features in the phase behaviour, with the helicoidal SmC formed from achiral molecules standing out as something of interest well beyond the liquid crystal field. I strongly recommend publication as is.

Minor comment: the domains in fig S6 are very hard to see. Could the grey scale be adjusted?

Reviewer #2 (Remarks to the Author):

The manuscript "Helicoidal smectic phases formed by achiral molecules" by J. P. Abberley et al. reports the synthesis and the characterization of a new homologous series of asymmetric and achiral bent-shaped mesogens. The synthesized compounds show rich phase diagrams, with a variety of

nematic and smectic phases, including several phases with heliconical arrangement of the achiral mesogenic molecules. These broken-chiral-symmetry phases include the twist-bend nematic (NTB), which has been recently discovered and is actively investigated in the last few years. Two smectic phases with the same spontaneous short-pitch heliconical structure has been reported for a first time. Moreover, the appearance of the NTB and the heliconical smectic phases in the phase diagrams of the same homologous series is discussed and the Authors suggest that the symmetry breaking in all the three phases may arise from the same mechanism which has been proposed for the NTB phase. A rich variety of experimental techniques is applied for characterizing the different liquid crystal phases, their phase transitions, the positional and orientational order of the phases and their optical and electro-optical properties.

In my opinion, the paper matches well the main criteria for publication in NCOMMS;

- The manuscript is very important to scientists in the field of liquid crystals (and in many other fields)
- The results are surely novel, with two new phases and original phase diagrams claimed and supported by strong evidence
- The results are technically sound and well discussed

I think that the paper represents a significant advance in the understanding of the symmetry-breaking phenomena and will stimulate new theoretical and experimental investigations in the already very active field of bent-shaped mesogenic molecules and their phases.

Therefore, I strongly recommend the publication of this work in Nature Communications after the Authors have considered the minor remarks listed below.

Minor remarks and suggestions:

1) The chiral symmetry breaking in liquid crystal phases was first reported in two seminal papers of H. Takezoe group concerning the smectic phases of bent-core molecules (Niori et al., *J. Mater. Chem.*, 6 (1996) 1231; Sekine et al., *Jpn. J. Appl. Phys.*, 36 (1997) 6455). I think that these papers should be cited and more detailed arguments should be given to prove that the SmCTB phase is different from the already reported phases.

By the way, I speculated in Ref. [4] that the elastic instability proposed for the NTB phase may also be involved in the chiral symmetry breaking in smectic phases of bent-shaped molecules, as an alternative to the polar in-layer interactions (“the nematic elasticity can give rise to a similar symmetry breaking, even in apolar banana smectics”). Also, I suggested that the NTB may be preempted by a smectic phase due to the similar scale of the NTB and smectic pitches (“It is also possible that a nematic-to-smectic transition precedes the sign change of K_3 , induced by the coupling of the director spatial modulation with the smectic mass density wave.”). In my knowledge, the present work gives the first experimental evidence supporting these speculations.

2) The discussion of the optical properties may be improved, by giving more details about the observation geometry and by more precise analysis of the observations. For example, the fact that no rotatory power (or optical activity in other terms) has been detected in the NTB and SmCTB phases does not imply that “the size of the chiral domains in these phases is smaller than the optical wavelength.”. Indeed, the optical activity here is not of molecular origin, it is structural (also called “extrinsic”). As in a cholesteric with small pitch, $P \ll \lambda$, the rotatory power decreases as P^3 : therefore, for so small pitch it will be undetectable. The real question is: why it is observed in the even shorter-pitch HexI phase? The answer is simple: as in this phase the pitch is about 2.2 layers, the optical axis rotates azimuthally at about 160° from layer to layer; optically, this is equivalent to rotation of the optical axis at -20° ; then, the optical period of the structure is 18 layers, and the rotatory power is about 100 times larger than in the NTB or SmCTB.

For the CD the situation is exactly the same – it is extrinsic, not due to the molecular chirality (for the N^* phase see, e.g., F.D. Saeva, J.J. Wysocki, JACS, 93 (1971) 5928). In fact, the CD is an exact analog of the rotatory power, with the imaginary part of the dielectric tensor replacing the real one.

In several places in the optical discussions there are some conclusions, not fully supported by the observations. For example, the statement “This optical uniaxiality (of the SmCTB phase) is inconsistent with a simple synclinic or anticlinic SmC phase structure, and indicates that averaging of molecular orientation must take place through the formation of a helioconical structure” disregards the possibility of a synclinic or anticlinic *chiral* SmC phases, as those observed in banana-like smectics.

3) The absence of a flexoelectric response is surprising, as in the right geometry it is a symmetry-defined property (see C. Meyer et al., PRL 111, 067801 (2013)) and should exist in any twisted structure (NTB, SmCTB, HexI), even if the molecules are achiral. It will be useful to give, e.g. in SI, the exact experimental conditions of the flexoelectric experiment, as a guideline for further studies.

Ivan Dozov

Reviewer #3 (Remarks to the Author):

The manuscript “Heliconical smectic phases formed by achiral molecules” by Abberley et al., reports the observation of a heliconical smectic phase with an extraordinary small pitch. The paper is well written and provides an essential contribution to the understanding of novel materials exhibiting self-assembled helical structures on a nano-scale. Some minor points about the text:

- The introduction could be a bit enhanced by discussing of spontaneously assembled helical structures in the broader perspective of the soft matter.
- Considering the variety of phase behaviour of the liquid crystals with N-Ntb-Smectic transition, the authors could also mention the formation of undulated smectic phases as described in Sebastian et al., PCCP, DOI: 10.1039/c6cp03899a and a similar discussion of a possible smectic-twist-bend structure by Tamba et al. in RSC Adv., DOI: 10.1039/c4ra14669g.
- Why do the authors claim that the smectic-C phase is the twist-bend type? Why isn't different from a helical SmC*? Is there particular experimental evidence?
- Are the Smectic-A phases of de-Vriese type?

Apart from that, I find the paper very interesting, accurately written. I suggest publication it in Nature Communications.

We thank the referees for their very favourable comments. Our responses to their specific comments are:

Reviewer #1 (Remarks to the Author):

- 1) Minor comment: the domains in fig S6 are very hard to see. Could the grey scale be adjusted?

The contrast and brightness of Fig. S6 have been adjusted as suggested, and now the chiral domains are more visible

Reviewer #2 (Remarks to the Author):

Minor remarks and suggestions:

- 1) The chiral symmetry breaking in liquid crystal phases was first reported in two seminal papers of H. Takezoe group concerning the smectic phases of bent-core molecules (Niori et al., J. Mater. Chem., 6 (1996) 1231; Sekine et al., Jpn. J. Appl. Phys., 36 (1997) 6455). I think that these papers should be cited and more detailed arguments should be given to prove that the SmCTB phase is different from the already reported phases.

In the Niori et al paper cited by referee, the ferroelectric properties of bent core mesogens were described for the first time but chiral symmetry breaking is not specifically mentioned in this article. The chiral properties (helical structure) of achiral bent core mesogens (B4 phase) were described by the same Tokyo group a year later, and therefore we have decided to cite the Sekine et al paper in the introduction (new reference 1) as suggested by the referee.

- 2) By the way, I speculated in Ref. [4] that the elastic instability proposed for the NTB phase may also be involved in the chiral symmetry breaking in smectic phases of bent-shaped molecules, as an alternative to the polar in-layer interactions (“the nematic elasticity can give rise to a similar symmetry breaking, even in apolar banana smectics”). Also, I suggested that the NTB may be preempted by a smectic phase due to the similar scale of the NTB and smectic pitches (“It is also possible that a nematic-to-smectic transition precedes the sign change of K3, induced by the coupling of the director spatial modulation with the smectic mass density wave.”). In my knowledge, the present work gives the first experimental evidence supporting these speculations.

We have added a statement that Dozov raised the possibility of the formation of helical smectic phases by achiral bent-core molecules (reference 5).

- 3) The discussion of the optical properties may be improved, by giving more details about the observation geometry and by more precise analysis of the observations. For example, the fact that no rotatory power (or optical activity in other terms) has been detected in the NTB and SmCTB phases does not imply that “the size of the chiral domains in these phases is smaller than the optical wavelength.”. Indeed, the optical activity here is not of molecular origin, it is structural (also called “extrinsic”). As in a cholesteric with small pitch, $P \ll \lambda$, the rotatory power decreases as P^3 : therefore, for so small pitch it will be undetectable. The real question is: why it is observed in the even shorter-pitch HexI phase? The answer is simple: as in this phase the pitch is about 2.2 layers, the optical axis rotates azimuthally at about 160°

from layer to layer; optically, this is equivalent to rotation of the optical axis at -20° ; then, the optical period of the structure is 18 layers, and the rotatory power is about 100 times larger than in the NTB or SmCTB. For the CD the situation is exactly the same – it is extrinsic, not due to the molecular chirality (for the N* phase see, e.g., F.D. Saeva, J.J. Wysocki, JACS, 93 (1971) 5928). In fact, the CD is an exact analog of the rotatory power, with the imaginary part of the dielectric tensor replacing the real one.

In our opinion the chiral optical effects are not related to the helical structure. Even if the helical pitch length is 18 smectic layers in the HexI phase, the ORP should be negligible because the pitch (~ 90 nm) is much shorter than the wavelength of the visible light. Therefore we believe the mechanism of OA could be similar to that in the B4 phase, i.e. the 'layer chirality'. This effect is related to the strongly hindered rotation of non-rod-like molecules. However, we would rather avoid further speculation as this hypothesis still requires additional verification.

- 4) In several places in the optical discussions there are some conclusions, not fully supported by the observations. For example, the statement "This optical uniaxiality (of the SmCTB phase) is inconsistent with a simple synclinic or anticlinic SmC phase structure, and indicates that averaging of molecular orientation must take place through the formation of a helioconical structure" disregards the possibility of a synclinic or anticlinic chiral SmC phases, as those observed in banana-like smectics.

On this matter we disagree with the referee. Synclinic or anticlinic SmC phases consisting of not only rod-like molecules but also banana-like molecules are in general biaxial. Only for a very special tilt angle and/or molecular bend, might the phase become uniaxial (for example, the orthoconic SmCA phase).

- 5) The absence of a flexoelectric response is surprising, as in the right geometry it is a symmetry-defined property (see C. Meyer et al., PRL 111, 067801 (2013)) and should exist in any twisted structure (NTB, SmCTB, HexI), even if the molecules are achiral. It will be useful to give, e.g. in SI, the exact experimental conditions of the flexoelectric experiment, as a guideline for further studies.

The set-up for the flexoelectric measurements was a conventional one, and the experimental conditions have been added to the SI as requested by the referee.

Reviewer #3 (Remarks to the Author):

Some minor points about the text:

1. The introduction could be a bit enhanced by discussing of spontaneously assembled helical structures in the broader perspective of the soft matter.

This is a communication type publication and so we prefer to avoid a lengthy introduction.

2. Considering the variety of phase behaviour of the liquid crystals with N-Ntb-Smectic transition, the authors could also mention the formation of undulated smectic phases as described in Sebastian et al., PCCP, DOI: 10.1039/c6cp03899a and a similar discussion of a possible smectic-twist-bend structure by Tamba et al. in RSC Adv., DOI: 10.1039/c4ra14669g.

Both papers quoted by the referee describe the phase behaviour of the same material. In the earlier paper there is the suggestion that the smectic phase seen below the NTB phase might be helical, but in the later publication the authors reported precise X-ray measurement and showed that the phase observed actually has a modulated broken layer structure (B1). Therefore we have decided to include only the latter paper by Sebastian et al. (new reference 16).

3. Why do the authors claim that the smectic-C phase is the twist-bend type? Why isn't different from a helical SmC*? Is there particular experimental evidence?

The SmC* designation is reserved for helical tilted smectic phases consisting of chiral molecules. To avoid any confusion, we prefer to retain the SmC_{TB} abbreviation to refer to a helical phase consisting of achiral bent molecules.

4. Are the Smectic-A phases of de-Vriese type?

We do not see any evidence for de-Vries type behaviour. Thus, there is a small but distinct decrease of the layer spacing at the SmA-SmC_{TB} phase transition and the cone angle values deduced from x-ray measurements agree well with those calculated from optical measurements.

REVIEWERS' COMMENTS:

Reviewer #2 (Remarks to the Author):

I am fully satisfied by the Authors' reply and the changes in the paper. I strongly recommend its publication in the present form.

Reviewer #3 (Remarks to the Author):

I believe, the paper is suitable for publication in the current form.

We thank the referees for their very favourable comments. Our responses to their specific comments are:

Reviewer #2 (Remarks to the Author):

I am fully satisfied by the Authors' reply and the changes in the paper. I strongly recommend its publication in the present form.

Reviewer #3 (Remarks to the Author):

I believe, the paper is suitable for publication in the current form.

We fully agree with both referees.